# Three-Dimensional Digital Image Analysis of Skeletal and Soft Tissue Points A and B after Orthodontic Treatment with Premolar Extraction in Bimaxillary Protrusive Patients

**DOI:** 10.3390/biology11030381

**Published:** 2022-02-27

**Authors:** You Na Lim, Byoung-Eun Yang, Soo-Hwan Byun, Sang-Min Yi, Sung-Woon On, In-Young Park

**Affiliations:** 1Division of Orthodontics, Hallym University Sacred Heart Hospital, Anyang 14066, Korea; 1004lun@hanmail.net; 2Division of Oral & Maxillofacial Surgery, Hallym University Sacred Heart Hospital, Anyang 14066, Korea; face@hallym.ac.kr (B.-E.Y.); purheit@daum.net (S.-H.B.); queen21c@gmail.com (S.-M.Y.); 3Graduate School of Clinical Dentistry, Institute of Clinical Dentistry, Hallym University, Chuncheon 24252, Korea; drummer0908@hanmail.net; 4Division of Oral & Maxillofacial Surgery, Hallym University Dongtan Sacred Heart Hospital, Hwaseong 18450, Korea

**Keywords:** malocclusion, Class I malocclusion, hard tissue, soft tissue changes, CBCT, three dimensional, digital

## Abstract

**Simple Summary:**

Malocclusion is a misalignment or inappropriate relationship between the upper and lower dental arches when the jaws close. Orthodontics, such as tooth extraction, clear aligners, or dental braces, are frequently used to address malocclusion, followed by growth modification in children or orthognathic surgery in adults. The treatment goals are to improve facial and dental esthetics, functional occlusion, periodontal health, and stability. It is also feasible to achieve an esthetic improvement of the soft tissue. This study shows how soft tissues change after extraction of premolars in patients with Angle Class I bimaxillary alveolar protrusion through three-dimensional analysis. The results show that changes in soft tissue point A and skeletal point A are three-dimensionally related.

**Abstract:**

*Aim.* To investigate the effect of changes in incisor tip, apex movement, and inclination on skeletal points A and B and characterize changes in skeletal points A and B to the soft tissue points A and B after incisor retraction in Angle Class I bimaxillary dentoalveolar protrusion. *Methods.* Twenty-two patients with Angle Class I bimaxillary dentoalveolar protrusion treated with four first premolar extractions were included in this study. The displacement of skeletal and soft tissue points A and B was measured using cone-beam computed tomography (CBCT) using a three-dimensional coordinate system. The movement of the upper and lower incisors was also measured using CBCT-synthesized lateral cephalograms. *Results.* Changes in the incisal tip, apex, and inclination after retraction did not significantly affect the position of points A and B in any direction (x, y, z). Linear regression analysis showed a statistically significant relationship between skeletal point A and soft tissue point A on the anteroposterior axis (z). Skeletal point A moved forward by 0.07 mm, and soft tissue point A moved forward by 0.38 mm, establishing a ratio of 0.18: 1 (r = 0.554, *p* < 0.01). *Conclusion.* The positional complexion of the skeletal points A and B was not directly influenced by changes in the incisor tip, apex, and inclination. Although the results suggest that soft tissue point A follows the anteroposterior position of skeletal point A, its clinical significance is suspected. Thus, hard and soft tissue analysis should be considered in treatment planning.

## 1. Introduction

The progression of digital imaging methods and tools has led to advancements in diagnosis and treatment planning and has played a prominent role in the field of dentistry [1]. In particular, the importance of three-dimensional (3D) imaging techniques has been increasingly recognized in some patients requiring the achievement of a harmonious soft tissue profile [2].

In Angle Class I bimaxillary dentoalveolar protrusion, retraction of the maxillary and mandibular incisors is required to position the incisors in a stable place within the oral cavity to relieve facial protrusion. In addition, the treatment that retracts incisors after extracting four maxillary and mandibular premolars is preferred to achieve an esthetically desirable soft tissue profile and lip incompetency [3]. Thus, it is crucial to study changes in the relationship between soft tissues and dentoalveolar structures that outline the treatment outcome of orthodontic tooth movement [4].

Points A and B have been commonly used to define the anteroposterior relationship of the maxilla and mandible, facilitated by nearly all popular analyses [5]. However, the two anatomical landmarks are affected by dentoalveolar bone remodeling with orthodontic treatment and growth [6,7]. Several studies have shown that incisal inclinations influence the position of points A and B. Al-Abdwani et al. showed that each 10-degree retroclination of incisors resulted in a statistically significant change of 0.4 and 0.3 mm in the horizontal plane at points A and B, respectively [8]. Hassan et al. found that each 10-degree retroclination of incisors resulted in a 0.6-mm displacement at point A superiorly [9]. However, there has been no evidence that changes in incisal inclination result in statistically significant positional displacement at point B.

Many previous studies have used two-dimensional (2D) analysis to quantify the facial soft tissues and focused on changes only in the midsagittal plane using lateral cephalograms [8,9,10,11]. Two-dimensional measurements have limited congruity and relevance when evaluating the 3D dentofacial complex [12,13]. The relationship between dentoalveolar movement and changes in soft tissue is complicated and varies in all three planes of space [14,15,16,17,18]. In addition to cone-beam computed tomography (CBCT) technology, multiple pieces of 3D software have enabled clinicians to plan and assess the treatment with a more comprehensive view of the dentofacial structures [19]. In the present study, we reconstructed the whole surface of skeletal and soft tissue from pre-treatment and post-treatment CBCT images. The measurements were conducted using advanced approaches with sophisticated software. Software tools automatically aligned 3D datasets and used color maps to compare changes in 3D landmarks after orthodontic treatment. To the best of our knowledge, no study has evaluated the positional changes in points A and B after incisor retraction in a 3D approach.

This study aimed to analyze the effect of changes in incisor tip, apex movement, and inclination on points A and B, and characterize changes in the skeletal points A and B to the soft tissue points A and B after incisor retraction in Angle Class I bimaxillary dentoalveolar protrusion patients.

## 2. Materials and Methods

This study was approved by the institutional review board of the Hallym University Sacred Heart Hospital (IRB approval No. 2021-09-006-003).

Complete enumeration was performed to select samples that fitted the study objectives and met inclusion criteria. This study included 22 adults (17 women and five men) diagnosed with Angle Class I bimaxillary dentoalveolar protrusion who underwent orthodontic treatment with extraction of the maxillary and mandibular first premolars between 2012 and 2020 at the Department of Orthodontics, Hallym University Sacred Heart Hospital. The number of participants was calculated by using G*Power software (version 3.1.9.7; Heinrich-Heine-Universität Düsseldorf, Düsseldorf, Germany) using a significance level of α = 0.05, 80% power, and an effect size of 0.50.

The inclusion criteria were the following: (1) minimum age of 18 years, (2) well-aligned arches without or with minor crowding, (3) availability of pre- and post-treatment CBCT data, (4) lack of history of systemic disease, and (5) no orthodontic treatment in the past.

This study was performed using pre-adjusted McLaughlin, Bennett, and Trevisi (MBT) appliances with 0.22-inch slots after the premolars’ extraction. All patients were treated with conventional anchorages, such as transpalatal arch (TPA) and elastic chains for space closure. Before and after treatment, CBCT (Alphard-Vega 3030, Asahi Roentgen Ind. Co., Ltd. Kyoto, Japan) images were obtained in centric occlusion under the conditions of 80 kV, 5 mA, 17-s exposure time.

For the pre-treatment (T0) and post-treatment (T1) CBCT images, a coordinate system with the Frankfort horizontal plane parallel to the xy-plane, the line connecting the orbitale parallel to the x-axis, and the pogonion being set as the zero point was followed. CBCT cephalograms were synthesized from the reoriented CBCT data, and linear measurements were performed using the OnDemand3D software (Cybermed, Seoul, Korea) (Figure 1). To determine the total distance of the incisor movement, a vertical reference line (vert T) constructed through a stable craniofacial structure was used to measure the distance between the incisal tip and root apex between T0 and T1 (Figure 2).

The Digital Imaging and Communications in Medicine (DICOM) files were converted into the. stl format, the standard file type representing the 3D surface geometry (Figure 3). Skeletal and soft tissue surfaces were constructed using the Invesalius open-access software (Renato Archer Information Technology Center, Campinas, Brazil). A specific density within the images that were derived from different shades of gray was customized by moving the threshold bars using an advanced 3D processing software (Geomagic Control X, 3D Systems, Rock Hill, SC, USA). An auto-alignment of the STereoLithography (STL) files obtained at T0 and T1 was performed, and the correspondence between the reference (T0) and measured (T1) data was checked (Figure 4).

Four reference points (points A, B, A, and B) were examined to compare changes at T0 and T1 (Figure 5 and Figure 6). Using the 3D compare function, a color deviation map between the reference and measured objects was created, and the selected deviation values were identified (Figure 4C). Each coordinate value was marked in accordance with the trigonal system (x, y, z) and recorded in the program. The x-axis, y-axis, and z-axis indicate the right and left, up and down, as well as anterior and posterior relationships, respectively.

To assess intra-observer reliability, all measurements were re-performed two weeks after the first measurements by the same examiner (LYN).

Statistical analyses were performed using the SPSS software (version 18; IBM, Armonk, NY, USA). The intraclass correlation coefficient (ICC) was calculated to indicate the reproducibility of the intra-examiner repetitive identification. The Shapiro–Wilk test was used in an attempt to evaluate whether the results followed a lognormal distribution. The coordinate values of T0 and T1 were evaluated using the paired t-test and Wilcoxon signed-rank test for paired samples with and without assumptions regarding distribution, respectively. Linear regression analysis was performed to assess the relationship of positional changes in points A and B with the incisal inclination, movement of the incisal tip and apex, and soft tissue points A and B. The significance level for all statistical analyses was <0.05.

## 3. Results

Out of 22 patients, 17 were women, and five were men. The mean age of the patients was 25.62 ± 7.77 years at the start of treatment. The ICC showed a mean of 0.93 (ICC, 0.86–0.98), indicating excellent reproducibility in the intra-examiner repeatability.

The results of changes in points A and B as measured using the 3D program between T0 to T1 are listed in Table 1.

At point A, the changes along the x-, y-, and z-axes between T0 and T1 were –0.49 ± 0.12 mm, 0.383 ± 0.51 mm, and 0.07 ± 0.25 mm, respectively. The changes were statistically significant along the y- and z-axes. At point B, the changes along the x-, y-, and z-axes were 0.01 ± 0.18 mm, 1.02 ± 0.80 mm, and −0.29 ± 0.25 mm, respectively. The changes were statistically significant along the z-axis.

Changes along the x-, y-, and z-axes between T0 and T1 were −0.11 ± 0.16 mm, 1.55 ± 1.02 mm, and 0.38 ± 0.28 mm in soft tissue point A and −0.01 ± 0.13 mm, 1.84 ± 1.47 mm, and −0.20 ± 0.54 mm in soft tissue point B, respectively. Changes in soft tissue point A were statistically significant for all axes. However, there was no significant difference between T0 and T1 in soft tissue point B.

The tip of the upper and lower incisors moved backward by 6.2 mm and 5.6 mm, respectively. Apices of the upper and lower incisors showed 0.2-mm and 0.6-mm retraction, respectively, following treatment. The incisor mandibula plane angle (IMPA) and upper incisor to sella nasion (SN-U1) angle decreased by 11.2° and 11.5°, respectively (Table 2).

However, the changes in the incisal tip, apex, and inclination after retraction did not show significant effects on the A and B positions in any direction (Table 3).

The results provide evidence that the displacement of point A along the z-axis results in statistically significant changes in soft tissue point A along the z-axis. The coefficient of determination (R2) was 0.307. On the other hand, there was no evidence that changes in point B resulted in significant positional changes in soft tissue point B in any direction (Table 4).

## 4. Discussion

Three-dimensional imaging technology has opened up new possibilities for orthodontic diagnosis and treatment evaluation. In both in vitro and in vivo studies conducted using dry skulls by Kumar et al. [20], cephalometric measurements from synthesized CBCT are not different from those of conventional cephalometric analyses. Park et al. identified significant differences in one linear (distance between U1 and facial plane) and three angular (gonial angle; ANB difference; and facial convexity) measurements; however, there were no significant differences between the conventional lateral and CBCT-synthesized cephalometric radiographs [21]. However, errors due to incorrect patient positioning during image acquisition can be corrected through iterative adjustments in the CBCT data set [22].

Previous studies have evaluated the changes only in the midsagittal area because they used conventional lateral cephalograms for their measurements [2,8,9,10,11]. In addition, considering the tracing error, it may not be suitable for detecting subtle changes with the conventional lateral cephalometric analysis. Therefore, the advantages of our method via advanced 3D software are quick configuration, surface-based registration with high accuracy, and measurements in all three planes of space [23]. The average of the positional changes of point A in the z-axis was positive, indicating that point A moved forward. This result may be due to the type of tooth movement and bone modeling during retraction. Some authors state that point A is a landmark, influenced by growth and dentoalveolar remodeling during orthodontic treatment [5,6,7,24]. Given that all patients had a minimum age of 18 years, it was presumed that the influence of growth on points A and B changes would be clinically negligible and have minor effects on the treatment results. Frost believed that mechanical compression related to bone formation and tension is related to resorption [25]. He suggested that apposition could be perceived as a response to the bending of the alveolar wall. Sarikaya et al. agreed that bone apposition or plastic deformation of the cortical plate occurs in the compression area [26].

In this study, the maxillary incisors moved 0.19 ± 2.06 mm to the rear at the apical height, while the incisors tip was retracted by 6.51 ± 2.18 mm on average, thereby showing a tendency of a tipping movement rather than a translation of the teeth. These results are consistent with those of Vardimon et al., who observed that the retraction of incisors with torque resulted in a combined movement with some tipping rather than bodily movement [27]. Goldin et al. reported that the upper incisor became more upright with the labial root torque producing a greater rate of advancement of point A [28]. Although the mean of the incisal changes indicated controlled tipping of the upper incisors, the number of cases with uncontrolled tipping during retraction was 9 out of 22 in this study. These cases mainly showed that the upper incisors tend to move forward at point A. Given that responses to stimuli are distributed regionally, it could be assumed that the forces concentrated in the apex of the alveolus during uncontrolled tipping stimulated bone apposition, and the adjacent skeletal point A was also affected [29].

The total change in the position of point B after orthodontic treatment was in the backward direction. However, considering the correlation of point B with the overlying soft tissue, no significant relationship was found between the hard and soft tissue changes in the sagittal direction. These results are slightly different from the findings of Hosseinzadeh-Nik et al. [30], where retraction of the anterior teeth led to retraction of point B, and changes in soft tissue point B were followed by changes in the corresponding point of the underlying hard tissue. There are two possible explanations for these results. It would not have been able to eliminate the errors due to mandibular rotation caused by changes in occlusion after treatment because superimposition was accomplished based on the constant points other than the mandibular symphysis [31]. In addition, there would be more chances to find a significant correlation between changes in the position of point B and soft tissue point B with a larger sample size.

In contrast, there was a statistically significant relationship between the retraction of skeletal point A and soft tissue point A following incisor retraction. Skeletal point A moved forward by 0.07 mm (*p* < 0.05), and soft tissue point A moved forward by 0.38 mm (*p* < 0.001), establishing a ratio of 0.18: 1 (r = 0.554, *p* < 0.01). Sharma found the ratio of point A to soft tissue point A to be 1.5: 1 (r = 0.648, *p* < 0.01) in Class I bimaxillary dentoalveolar protrusion cases [11], which differed from those other studies. Many authors stated that the amount of change in the skeletal profile varies from subject to subject, and changes in the hard tissues are not always reflected by equivalent changes in the overlying soft tissues [32,33]. Our results might be considered clinically irrelevant because changes in the skeletal and soft tissue point A did not have sufficient magnitude to generalize the ratio, and the R^2^ was 0.307. This means that skeletal point A does not account for 70% of the variation in soft tissue point A along the z-axis. It would be an indicator of a lower fit for the observations.

For treatments in which improving the soft tissue profile is more important, such as treatment for bimaxillary dentoalveolar protrusion, analyzing the skeletal relationship based on the correlation between the amount of change in soft and hard tissue before and after treatment is relevant. It is believed that changes in the soft tissues result from changes in the hard and soft tissue tension, thickness, length, or other factors, such as fat and muscle components. Therefore, to predict changes in the soft tissue, changes in the hard tissue and other factors must be considered together. Because changes in soft tissue occur continuously in adults, and soft tissue shows slower adaptation than hard tissue, it is thought that continuous research on changes in soft tissue during the retention period is necessary [34].

Another clinical relevance of this study is that the root apex, incisal tip, and inclination do not determine the position of skeletal points A and B. The bodily retraction would bring about the proximity of the root apex to the palatal cortex plate and abundant alveolar bone resorption [35]. Therefore, bodily retraction of considerable distance is not ideal for the treatment of bimaxillary dentoalveolar protrusion, and careful tooth movement is necessary, considering the location of the tooth within the alveolar bone [30,36]. In other words, clinicians should place the anterior teeth in the most esthetic position through initial uprighting and some translation rather than focusing on the retraction of skeletal points A and B.

## 5. Conclusions

In the z-axis direction, changes in skeletal point A led to changes in the overlying corresponding soft tissue point A. However, the magnitude of change was not large enough to be clinically relevant. Changes in the incisal tip, root apex, and inclination were not significantly correlated with changes in the position of skeletal points A and B in any direction. Thus, changes in points A and B seem complex and have multifactorial consequences that cannot be attributed to any single factor.

Based on the dental and skeletal analysis, it is inappropriate to estimate the treatment plans and outcomes in bimaxillary dentoalveolar protrusion. Thus, soft tissue analysis, closely related to facial esthetics, should be performed with dental and skeletal analysis.

## Figures and Tables

**Figure 1 biology-11-00381-f001:**
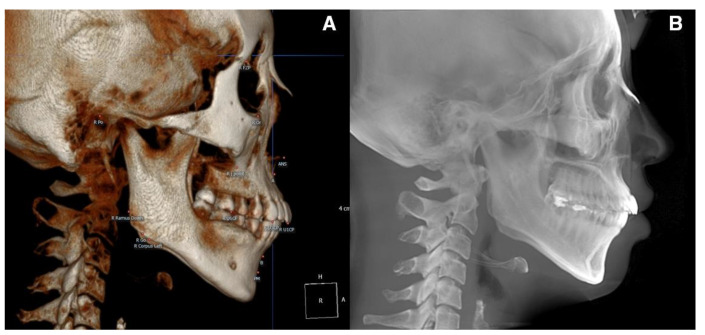
(**A**), CBCT 3D image view. The image was oriented along the Frankfort horizontal plane in reference to the right porion, right orbitale, and left orbitale. (**B**), CBCT-synthesized lateral cephalogram was constructed in accordance with orientation.

**Figure 2 biology-11-00381-f002:**
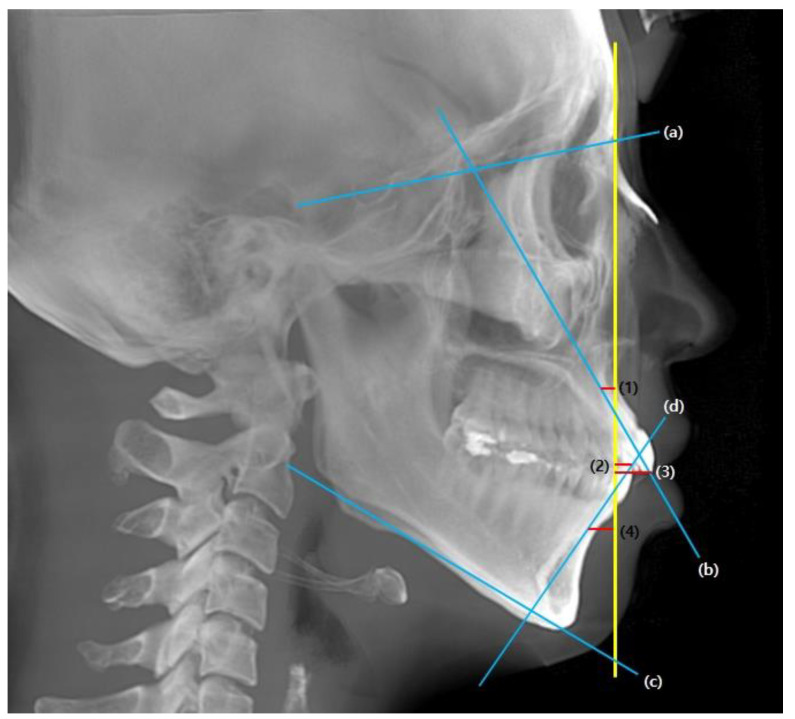
Cephalometric reference planes and measurements. Yellow, vertical reference line (vert T); Blue, (**a**) S-N (**b**) U1-SN (**c**) mandibular plane (**d**) IMPA; Red, (1) U1 apex (2) L1 tip (3) U1 tip (4) L1 apex. Incisal movements were quantified by measuring the horizontal distance from incisor tip and apex to vert T, which is Nasion perpendicular line.

**Figure 3 biology-11-00381-f003:**
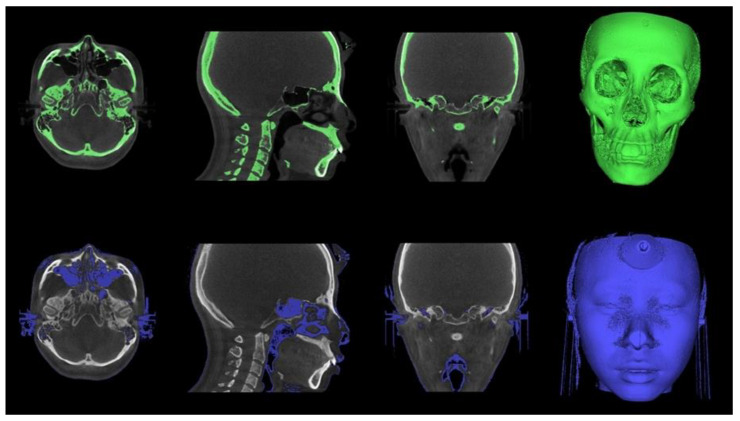
Converting Dicom file to STL file with InVesalius software. The specific threshold value was applied to segment between the skull structures and soft tissues.

**Figure 4 biology-11-00381-f004:**
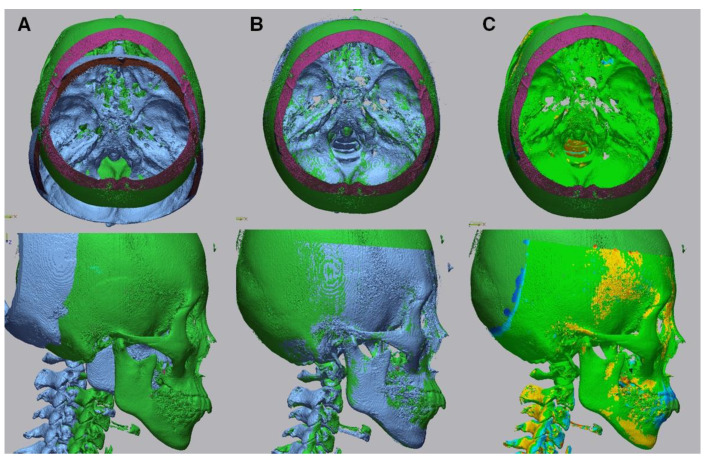
Alignment of the pre-treatment and post-treatment surface data. (**A**), Before superimposition. (**B**), The superimposition process involved the following two steps; (i) initial alignment was performed before running other alignment types. The software automatically calculated an initial fit between the two objects to get the Reference (pre) and Measured (post) data close and register features. (ii) Best fit alignment was then applied to calculate the best fit between the two objects automatically. (**C**), 3D comparison function after alignment.

**Figure 5 biology-11-00381-f005:**
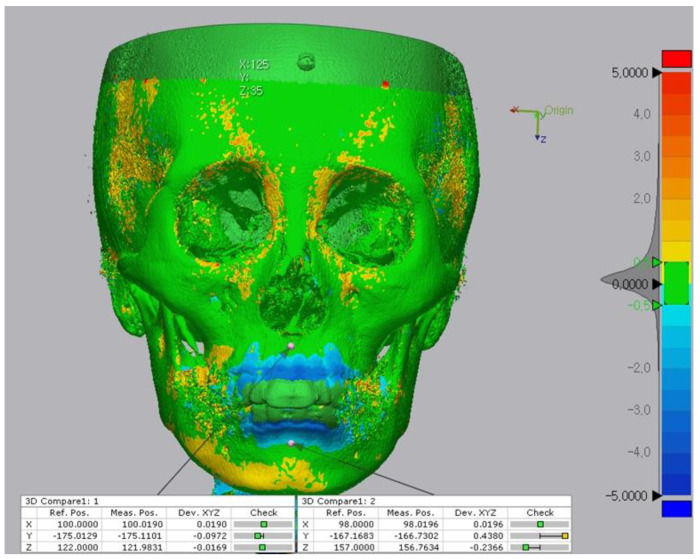
Comparison of reference points (pre) and measure points (post) located on the bone surface after the superposition. The 3D discrepancies were analyzed on the basis of a color-coded map. The color scale ranges from −5 mm (blue), indicating subtractive changes to +5 mm (red), indicating additive changes. Green indicates no difference.

**Figure 6 biology-11-00381-f006:**
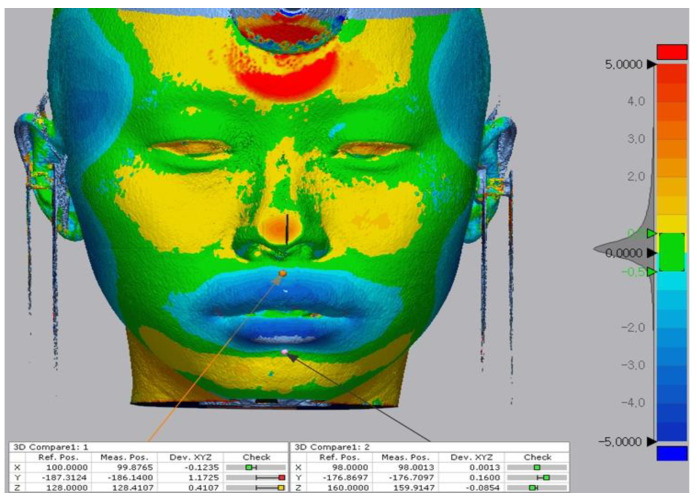
Comparison of reference points (pre) and measure points (post) located on the soft-tissue surface after the superposition. The 3D discrepancies were analyzed based on a color-coded map. The color scale ranges from −5 mm (blue) indicating subtractive changes to +5 mm (red) indicating additive changes. Green indicates no difference.

**Table 1 biology-11-00381-t001:** Mean changes of skeletal and soft tissue points A and B in three directions (T1–T0).

	Difference (SD)	Interquartile Range	*p*-Value
Skeletal point A			
Dx (mm)	−0.49 (0.12)	0.12	NS
Dy (mm)	0.38 (0.51)	0.69	†
Dz (mm)	0.07 (0.25)	0.1	*
Skeletal point B			
Dx (mm)	0.01(0.18)	0.09	NS
Dy (mm)	1.01 (0.80)	1.09	NS
Dz (mm)	−0.29 (0.25)	0.46	‡
Soft tissue point A			
Dx (mm)	−0.11 (0.16)	0.2	†
Dy (mm)	1.55 (1.02)	1.58	‡
Dz (mm)	0.38 (0.28)	0.41	‡
Soft tissue point B			
Dx (mm)	−0.01 (0.13)	0.16	NS
Dy (mm)	1.84 (1.47)	2.13	NS
Dz (mm)	−0.20 (0.54)	0.43	NS

* *p* < 0.05, † *p* < 0.01, ‡ *p* < 0.001. D, direction; Difference, mean changes (post-pre); NS, not significant; x, left and right relationship. Positive and negative values represent left and right movements, respectively; and y, up and down. Positive and negative values represent up and down movements, respectively; z, anterior, and posterior. Positive and negative values represent the anterior and posterior movements, respectively.

**Table 2 biology-11-00381-t002:** Mean cephalometric changes after treatment (T1–T0).

	Difference (SD)	Interquartile Range	Minimum	Maximum
SN-U1(°)	−11.51 (7.70)	11.84	−28.75	−1
IMPA (°)	−11.20 (5.98)	5.96	−25.75	0.8
U1 tip(mm)	−6.15 (2.18)	3.34	−10.2	−2.9
U1 apex (mm)	−0.19 (2.06)	2.75	−3.7	3.25
L1 tip (mm)	−5.59 (2.19)	2.96	−11.45	−2.45
L1 apex (mm)	−0.61 (2.22)	2.74	−4.65	2.95

mm, millimeters; °, degree; SN, sella-nasion; U1 tip; incisal tip of maxillary incisor, U1 apex; root apex of maxillary incisor.

**Table 3 biology-11-00381-t003:** The coefficient of correlation and *p*-values between incisal movements and skeletal points A and B changes.

		SN-U1	U1 tip	U1 apex
Skeletal point A				
Dx	R^2^	0.001	0.034	0
	P	NS	NS	NS
Dy	R^2^	0.023	0.025	0.076
	P	NS	NS	NS
Dz	R^2^	0	0.003	0.152
	P	NS	NS	NS
		IMPA	L1 tip	L1 apex
Skeletal Point B				
Dx	R^2^	0.061	0.065	0.013
	P	NS	NS	NS
Dy	R^2^	0.003	0.045	0.039
	P	NS	NS	NS
Dz	R^2^	0.041	0.001	0
	P	NS	NS	NS

D, deletion; R, coefficient of correlation; NS, not significant; SN, sella-nasion; U1 tip, incisal tip of maxillary incisor; U1 apex, root apex of maxillary incisor; IMPA, L1-mandibular plane; L1 tip, incisal tip of mandibular incisor; L1 apex, root apex of mandibular incisor.

**Table 4 biology-11-00381-t004:** The coefficient of correlation and *p*-values between skeletal and soft tissue points A and B changes.

		Skeletal point A		
		Dx	Dy	Dz
Soft tissue point A				
Dx	R^2^	0.135	0.010	0.052
	P	NS	NS	NS
Dy	R^2^	0.295	0.067	0.019
	P	NS	NS	NS
Dz	R^2^	0.001	0.041	0.307
	P	NS	NS	†
		Skeletal point B		
		Dx	Dy	Dz
Soft tissue point B				
Dx	R^2^	0.005	0	0.009
	P	NS	NS	NS
Dy	R^2^	0.051	0.002	0.002
	P	NS	NS	NS
Dz	R^2^	0.028	0.002	0.023
	P	NS	NS	NS

*p* < 0.05, † *p* < 0.01, *p* < 0.001. D, direction; R, coefficient of correlation; NS, not significant.

## Data Availability

The data that support the findings of this study are available from the corresponding author upon reasonable request.

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
