# Peer review of "Three-Dimensional Digital Image Analysis of Skeletal and Soft Tissue Points A and B after Orthodontic Treatment with Premolar Extraction in Bimaxillary Protrusive Patients"

_biology, 2022, doi:10.3390/biology11030381_

Round 1
Reviewer 1 Report
Please provide sample size calculation Please discuss significance of results, all sentences reported in the discussion should be sustained by methods and results, do not make unsustained claims Please asess data distribution prior to statistical analysis (I would appreciate raw output of SPSS software) Please give a summary table of descriptive statistics please discuss low r2 and significance of data
1. What is the main question addressed by the research?
2. Do you consider the topic original or relevant in the field, and if
so, why? I think that the subject is relevant yet its presentation lacks in clarity and data synthesis
3. What does it add to the subject area compared with other published
material? I think it does not add much to the available evidence however I believe it is one of the first studies addressing the issue of soft tissues and hard tissues changes after extraction treatment performed on CBCT scans 4. What specific improvements could the authors consider regarding the
methodology? Report sample size calculation Assessment of normality through shapiro wilk test Improve data synthesis, reporting mean and standard deviation or Md and IQR depending on data distribution for all assessed variables
5. Are the conclusions consistent with the evidence and arguments
presented and do they address the main question posed?
Discussion and conclusion lack in the explanation of the relevance of retrieved data that is: why the data are clinically relevant? what do the data add to the available knowledge? is statistical significance relevant to the clinical issues? 6. Are the references appropriate?
I do not have observation for the references in this first round of revision, they appear ok for now
Reviewer 2 Report
Dear Authors
the manuscript is interesting but some minor changes are necessary before taking it into consideration for publication. Please address the following concerns:
- specify what premolars have been extracted (first or second premolars?);
- was any difference in the soft tissue profile if the premolars extracted were the first or the second?
- Best regards
“Three-dimensional digital image analysis of skeletal and soft tissue points A and B after orthodontic treatment with premolar extraction in bimaxillary protrusive patients” the following observations were made:
> 1. What is the main question addressed by the research? YES.
> 2. Do you consider the topic original or relevant in the field, and if so, why? The topic is not original. Its importance is average for the orthodontic community.
> 3. What does it add to the subject area compared with other published material? It add a description of how points A and B change thieir positions after premolars extractions but Authors don’t specify what premolars have been extracted and the sample is of only 22 patients
> 4. What specific improvements could the authors consider regarding the methodology?
ABSTRACT
- For a better understanding of the abstract, the following subtitles should be presented: aim, materials and methods, results and conclusion.
- The conclusion should be better explained as point 5 (conclusion) of the content in this article.
MATERIALS AND METHODS
- According to the article “This study included 22 adults (17 women and five men) diagnosed with Angle Class I bimaxillary dentoalveolar protrusion who underwent orthodontic treatment with extraction of the maxillary and mandibular premolars between 2012 and 2020 at the Department of Orthodontics of the Hallym University Sacred Heart Hospital. “ but the sample of 22 people in 8 years of selection is enough to generalize the results? Please explain.
Please specify what premolars have been extracted (first or second) and if there is any difference between them in the obtained results.
> 5. Are the conclusions consistent with the evidence and arguments presented and do they address the main question posed? Yes. The conclusions are well described and in accordace with the topic of the article.
> 6. Are the references appropriate? Yes.
> 7. Please include any additional comments on the tables and figures.
Figures are rapresentative and legends well described.
- According to the article “ The significance level for all statistical analyses was less than 05.” but in table 1 of results different values of p are observed (see red arrow).
Round 2
Reviewer 1 Report
The authors' reply to my comments is: "please see the attachment". However no attachment is present in the dedicated section. Please provide me with the point by point response to the issues I have raised in the previous round of revision
Author Response
I am sorry for your inconvenience. The file that I attached was missed. Please see the attachment.

Round 3
Reviewer 1 Report
I believe that this paper is now fit for publication. Nonetheless I suggest to give a priori sample size calculation that gives information about the significance of data and the statisyical power of the study. https://www.graphpad.com/guides/prism/latest/statistics/stat_overview_of_sample_size_determ.htm
You may find at the link above some information and bibliography you may use to perform power analysis and sample size determination
